# Allometric Scaling Approaches for Predicting Human Pharmacokinetic of a Locked Nucleic Acid Oligonucleotide Targeting Cancer-Associated miR-221

**DOI:** 10.3390/cancers12010027

**Published:** 2019-12-19

**Authors:** Maria Teresa Di Martino, Mariamena Arbitrio, Massimiliano Fonsi, Claudio Alberto Erratico, Francesca Scionti, Daniele Caracciolo, Pierosandro Tagliaferri, Pierfrancesco Tassone

**Affiliations:** 1Department of Experimental and Clinical Medicine, Magna Graecia University, 88100 Catanzaro, Italy; scionti@unicz.it (F.S.); mercury86p@gmail.com (D.C.); tagliaferri@unicz.it (P.T.); 2Consiglio Nazionale delle Ricerche (CNR)–Istituto per la Ricerca e l’Innovazione Biomedica (IRIB)–Section of Catanzaro, 88100 Catanzaro, Italy; mariamenaarbitrio@yahoo.it; 3Citoxlab France, a Charles River Company, 27005 Evreux, CEDEX, France; Massimiliano.Fonsi@fr.citoxlab.com (M.F.); Claudio-Alberto.Erratico@fr.citoxlab.com (C.A.E.); 4College of Science and Technology, Temple University, Philadelphia, PA 19122, USA

**Keywords:** LNA-i-miR-221, phosphorothioate, oligonucleotide, pharmacokinetic, PK, plasma protein binding, locked nucleic acid, first-in-human, allometric method

## Abstract

LNA-i-miR-221 is a novel phosphorothioate backbone 13-mer locked nucleic acid oligonucleotide-targeting microRNA-221 designed for the treatment of human malignancies. To understand the pharmacokinetic properties of this new agent, including unbound/total clearance, we investigated the LNA-i-miR-221 protein binding in three different species, including rat (Sprague–Dawley), monkey (Cynomolgus), and human. To this end, we generated a suitable ultrafiltration method to study the binding of LNA-i-miR-221 to plasma proteins. We identified that the fraction of LNA-i-miR-221 (at concentration of 1 and 10 µM) bound to rat, monkey, and human plasma proteins was high and ranged from 98.2 to 99.05%. This high protein binding of LNA-i-miR-221 to plasma proteins in all the species tested translates into a pharmacokinetic advantage by preventing rapid renal clearance. The integration of these results into multiple allometric interspecies scaling methods was then used to draw inferences about LNA-i-miR-221 pharmacokinetics in humans, thereby providing a framework for definition of safe starting and escalation doses and moving towards a first human clinical trial of LNA-i-miR-221.

## 1. Introduction

A rising body of evidence indicates that microRNAs (miRNAs) can provide valuable therapeutic targets because of their potential to functionally regulate key oncogenic/tumor suppressor genes by simultaneous regulation of multiple-related pathways. Among several miRNAs, miR-221 has been widely investigated for its steady overexpression in a variety of solid and hematologic malignancies [1]. With the aim of miR-221 therapeutic targeting, a locked nucleic acid (LNA)-i-miR-221, a phosphorothioate 13-mer oligonucleotide (PS-ODN), was generated for specific miRNA inhibition. LNA-i-miR-221 has the advantages of both LNA technology and its PS backbone, resulting in increased seed sequence binding affinity and in vivo nuclease resistance [2]. LNA-i-miR-221 is an effective agent for targeting miR-221, upregulates canonical targets [3,4], induces significant anti-tumor activity against multiple myeloma (MM) and other malignancies [5], and rescues tumor sensitivity to alkylating agents [6]. On these bases, LNA-i-miR-221 has been selected and recently approved for a dose-escalation phase I clinical trial in humans (2017). However, at present, little information is yet available on the pharmacokinetics (PK) of LNA-oligonucleotides after systemic injection. In our previous studies on LNA-i-miR-221 PK properties in different animal species [2,4], we found that it reaches the targeted biphase already described for these “second generation” antisense oligonucleotides (ASOs), which relies on their systemic distribution and retention by tissues. Such processes involve surface protein interactions and endocytosis, which finally lead to cell internalization and excretion [7].

We have already developed a full GLP-compliant ion-pair reversed-phase LC–MS/MS method for accurate quantification of LNA-i-miR-221 in preclinical animal models that was recently validated for a GLP rat study [8]. This LNA-i-miR-221 detection method was applied to analysis of plasma samples from both preliminary toxicity studies in mice and monkeys [2], and for regulatory toxicity study in rats [4]. All the PK studies performed showed similar profiles in the tested species, showing a large systemic volume of distribution and extensive tissue penetration [2,4], supporting the results reported for other ASOs with a PS backbone [9]. Moreover, LNA-i-miR-221 showed rapid tissue distribution, followed by systemic clearance with low renal excretion [2], suggesting a favorable clearance reduction due to large binding to plasmatic proteins.

The aim of safe animal dose extrapolation is to estimate appropriate human dose and exposure levels to ensure drug safety. This approach is required for the initial dose selection in the clinical phase. In cross-species extrapolation, several factors should be considered in order to limit prediction bias. These include pharmacological, physiological, and anatomical factors, and metabolic function [10]. The relationship between the metabolic rate of an animal and its size is the basis of the allometric scaling approach. There is an inversely proportional relationship between these parameters. Body size is important in the rate of distribution of compounds for the correct conversion and calculation of the human equivalent dose (HED) [11]. Clearance (CL) and bioavailability are two important PK parameters directly related to first-in-human dose calculation [12]. Accurate predictions of human PK prior to phase I studies should help to determine the appropriate dosing regimen and to minimize safety risks in participants [13]. The sensitivity of organ clearance of a drug to changes in binding within blood depends on the drug’s unbound clearance. If unbound clearance is low relative to organ blood flow, the extraction ratio and CL will also be low and dependent on plasma binding [14]. Consequently, the measurement of the LNA-i-miR-221 free fraction in plasma may provide additional species comparison details that will be useful for estimating the unbound exposure and clearance in humans. To estimate the unbound exposure and CL of LNA-i-miR-221 in humans, we used an allometric interspecies scaling approach based on the plasma PK profile after intravenous (i.v.) administration of LNA-i-miR-221 [4]. Our approach took into account species differences in PK parameters (total CL and volume of distribution (Vd), as well as unbound drug CL and unbound Vd) via correction for the respective plasma protein binding. Due to the impact of this parameter on the rate of renal clearance via glomerular filtration, which is one of the major clearance mechanisms for LNA-i-miR-221, the integration of plasma protein binding in PK analysis could improve scaling accuracy. In fact, the calculation of unbound clearance may improve the predictions of the total clearance of ODNs in humans when the binding to plasmatic proteins is species-specific. Here, we describe the development of a suitable ultrafiltration method to determine the binding of LNA-i-miR-221 to human, monkey, and rat plasma proteins, and how these results can be integrated into multiple allometric interspecies scaling approaches in support of predicting a safe LNA-i-miR-221 dose in a first-in-human study.

## 2. Results and Discussion

### 2.1. Pharmacokinetic Analysis

Non-compartmental analysis methods (Phoenix WinNonlin software, v6.3: Pharsight Corporation, Mountain View, CA, USA) were used for PK characterization of the plasma concentration data relative to i.v. administration to mice, rats, and monkeys. Terminal first-order elimination rates for LNA-i-miR-221 from plasma (lambda z), were calculated using non-compartmental log-linear regression of the decay curves for the respective species. Half-life (t1/2) was calculated by dividing 0.693 by the first-order elimination rate. Terminal Vd values were calculated by dividing the calculated systemic CL by the terminal half-life value. All these results are summarized in Table 1. Figure 1, Figure 2 and Figure 3 show plasma concentrations of LNA-i-miR-221 (ng/mL)–time (h) profiles following a single IV bolus administration in male Sprague–Dawley rat, Cynomolgus monkey and NOD.SCID mice. Following i.v. administration, plasma concentrations rapidly declined in a multi-exponential fashion, characterized by a dominant initial rapid distribution phase wherein drug transferred from circulation to tissues in a few hours [2], followed by a much slower terminal elimination phase. The apparent terminal elimination rate observed in plasma was consistent with the slow elimination of LNA-i-miR-221 from tissues, indicating equilibrium between post-distribution phase plasma concentrations and tissue concentrations.

### 2.2. Allometric Scaling of Clearance Values

Cross-species regression using allometry was performed by linear regression of plasma clearance versus body weight (W). The equation used to relate the PK parameters (Y) to body weight (W) was as follows:Y = aW^b^(1a)

Estimations were made for both total (CLp) and unbound clearance (CLp^u^); the latter was calculated by correcting the measured CLvalues for the species-specific plasma protein-free fraction. Thus, a log-log regression of a PK parameter Y (i.e., total CLp or unbound CLp^u^ versus W) will yield a y-intercept value of “a” and a slope of “b”: logY = log(a) + b·logW.(1b)

The size and magnitude of exponent (b) indicates how the physiological variable changes as a function of body weight [12]. CL estimates were adjusted by mean body weight for mice, rats, monkeys, and humans using the values indicated in Table 2. These values have not been adjusted for the real animal weights [15]. The CL values, calculated via different allometric approaches, were compared and discussed in terms of consistency. The geometric mean of all the different calculated human PK parameters was adopted for the final estimation of CL and exposure.

### 2.3. Calculation of Human Equivalent Dose (HED)

According to the “Guidelines on strategies to identify and mitigate risk for first-in-human and early clinical trials with investigational medicinal products” (EMEA/CHMP/SWP/28367/07) and identification of the no observed adverse effect level (NOAEL), the AUC at the expected safe/effective dose in humans was simulated based on the dose calculated by directly applying the NOAEL dose in rats (5 mg/kg/day) and the dose normalized by body surface area, calculated by applying a correction factor using Equation (2), which finally predicted a HED dose of 0.78 mg/kg/day.
Correction factor = (human weight/animal weight)^(1 − 0.67)^(2)

As summarized in Table 1, i.v. bolus administrations in preclinical species have shown that the plasma concentration–time profile decreases rapidly in a polyphasic manner for mice, rats, and monkeys, with a rapid initial distribution phase in tissues and a much longer terminal phase. The initial t_1/2_ after i.v. bolus administration was relatively fast, generally within 1–1.5 h. For mouse PK, the first blood sample was collected 1.5 h after dosing and the initial rapid distribution phase was not accurately profiled. The extrapolated C_0_ value may therefore be less accurate in the mouse PK analysis. Additionally, he contribution of the initial distribution phase to the total AUC could not be correctly estimated. For this reason, the calculated mouse PK parameters were not used for the allometric scaling of human PK. The unbound clearance value was extrapolated from the total clearance using the following relationship:CLp^u^ = CLp/f_u_
where CLp^u^ is the unbound plasma clearance, f_u_ is the unbound fraction and CLp the total plasma clearance.

The unbound clearance in humans was then extrapolated by direct allometric scaling of the calculated unbound CL values for the preclinical species.

### 2.4. Allometric Scaling of PK Parameters

PK scaling was conducted using one or two species. The body weight values for human and preclinical species used in this prediction are reported in Table 2.

### 2.5. Two-Species Scaling

The following two methods were used to predict human drug CL (the equation is valid for both CLp and CLp^u^). The term CL is generically used to refer to total or unbound drug concentration.

(i) Scaling of CL from two species was based on Equation (1a), in which the CLs of both species were plotted against body weight (W). Direct scaling of total CLp from two species (rat and monkey) is shown in Table 3.

(ii) Tang et al. [16] suggested the following equations to predict human CL using rat and monkey data:
CL human (ml/kg) = a_rat–monkey_ × (W_human_)^0.65^(3)
where a_rat–monkey_ has the same value as parameter “a” (as calculated in Equation (1a)) and “b” is fixed at 0.65.

Scaling according to Tang et al. (as in Equation (3)) from rat and monkey CL is shown in Table 4.

For the prediction of human CLp using two-species scaling (option i), Equation (1a) was log transformed as Equation (1b).

The CL value logarithm in the preclinical species was then plotted against the body weight logarithm, and the linear fit of the transformed data was used to extrapolate the values of parameters “a” and “b” in Equation (1a). The human CL was then estimated using Equation (1a) and the calculated “a” and “b” parameters. In two-species scaling according to Tang et al. [16] (option ii), the “b” value was fixed at 0.65, while a_rat–monkey_ was calculated as discussed above.

### 2.6. One-Species Scaling

The following method was used to predict human drug CL:CL in human = Animal CL × (W_human_/W_animal_)^0.75^(4)

For the prediction of human total plasma CL using one-species scaling, the extrapolation of human CLp was calculated according to Equation (4). A distinct CL prediction was obtained from each preclinical species (rat and monkey; Table 5).

### 2.7. Plasma Protein Binding and CLp^u^

A suitable ultrafiltration method for determination of the binding of LNA-i-miR-221 to human, monkey, and rat plasma proteins was developed at Citoxlab France (Charles River Evreux, France). A previously validated LC/MS method for the quantification of LNA-i-miR-221 in rat plasma [8] was transferred and validated using an Orbitrap Q-Exactive (ThermoFisher). The following parameters were tested:(i)specificity, linearity, range and lower limit of quantification (LOQ),(ii)within-batch precision and accuracy,(iii)matrix effect,(iv)stability in human, monkey, and rat plasma at 37 °C.

The sample preparation was slightly adapted and optimized to the use of a high-resolution mass spectrometer [8] and the method was then successfully qualified. The rapid equilibrium dialysis device was originally chosen as an approach to determine the unbound protein fraction in this study, but the preliminary results using this device in the absence of proteins showed that the LNA-i-miR-221 was not able to freely pass through the semi-permeable membrane and reach equilibrium in the two chambers. Therefore, this approach was abandoned in favor of ultrafiltration. A coating step using a custom LNA–ASO analogue oligonucleotide to minimize unspecific absorption on the plastic surface was required to allow acceptable recovery rates. Using the ultrafiltration protocol, the overall fraction of the LNA-i-miR-221 (at concentration of 1 and 10 µM) bound to human, monkey, and rat plasma ranged between 98.2% and 99.05%. This result suggests that LNA-i-miR-221 is similarly highly bound to the plasma proteins of the three species tested (as described below and shown in Table 6 and Table 7).

### 2.8. Protein Binding

The unspecific binding was then assessed using an Amicon ultracentrifugation device. The results clearly showed high unspecific binding of the test item: 92% and 40% at 1 and 10 µM concentration, respectively (data not shown).

To limit the unspecific binding of LNA-i-miR-221 to the Amicon vials, they were pre-coated using a customized miRNA of similar length. The unspecific binding dropped to 50% and 35% at 1 and 10 µM, respectively (data not shown). It was then decided to use this coating step to assess the protein binding of the LNA-i-miR-221 and to apply a correction factor to compensate for the reduced free LNA-i-miR-221 recovery [17].

The fraction of the LNA-i-miR-221 bound to human, monkey, and rat plasma ranged between 98.2% and 99.05% (Table 8). The very low standard deviation values demonstrate the high reproducibility of the results obtained. Overall, our data suggested that LNA-i-miR-221 is highly bound to the plasma proteins of the three species tested, and that this high protein binding occurs similarly at both tested concentrations and in the three species.

The percentage of bound drug for propranolol, which is used as a reference control for highly bound drugs in human plasma, was 89.4% ± 0.75% in human plasma (*n* = 3). This result is in good agreement with the range of bound values expected and demonstrates the reliability of the protocol used.

The plasma protein binding value was used to calculate the unbound CL in each species as reported by Yu et al. [18]. The prediction study was further refined by interspecies comparison of the PS–ODN-free fraction in plasma, thus allowing estimation of the unbound exposure and clearance in humans.

The aim of this investigation was to develop a PK model to predict the CL and exposure of the oligonucleotide LNA-i-miR-221 in humans and to anticipate human plasma levels in the absence of other human data. Quantitative modeling approaches based on allometry for oligonucleotides have provided encouraging results, but may not be always straightforward [19]. For this reason, in our approach, different scaling methods were used in parallel, based on the available preclinical PK information, and the mean of the different estimates was finally used to predict the CL of LNA-i-miR-221 (and its exposure) in humans. The FDA Guidance for Industry [20] suggests that this approach can present a number of difficulties for estimation of a safe starting dose. Generally, at the time when an application to the competent regulatory authorities is made for an investigational new drug, there are a number of unknowns regarding animal toxicity and the comparability of human and animal PKs and metabolism:

(i) human metabolism may differ significantly from that of animals;

(ii) mechanisms of toxicity may not be known (e.g., toxic accumulation in a peripheral compartment);

(iii) toxicity may be due to an unidentified metabolite, not the parent drug.

Therefore, relying on PK models (based on the parent drug in plasma) to gauge starting doses would require multiple untested assumptions. In addition, uncertainties (such as differences between human and animal receptor sensitivity or density) have been shown to affect human pharmacological or toxicological outcomes.

Even though the monkey is considered to be the most appropriate species because monkeys experience the same dose-limiting toxicity as humans, for nonclinical safety assessment of LNA-i-miR-221, species selection was based on the retention of the miR-221 sequence among different species [21]. We then selected rats as the relevant species for the formal GLP toxicokinetic study, as it is a rodent species accepted by regulatory authorities. Specifically, for our study, we selected the Sprague–Dawley strain, as background data from previous studies were available at Citoxlab (France) where we performed the study.

PS–ODNs show similar PK profiles in plasma and tissue among species independently of sequence [22,23,24,25], with rapid CL from plasma, extensive tissue distribution [23,26,27] and a shorter half-life [22,23,28]. Our PK data showed the same trend as these observations and high binding to plasma proteins across the species.

The inclusion of a plasma protein binding correction in the allometric approaches adopted did not significantly modify the prediction. This is clearly related to the very similar unbound fraction measured ex vivo (ultrafiltration) in rat, monkey, and human plasma. The fact that all the tested species showed a similar unbound fraction at both concentrations (1 and 10 µM) tested in the protein binding assay is encouraging in terms of expected similar PK behavior across the species, which is a basic assumption when allometric extrapolation is applied to predict CL.

The HED dose and the relative exposure in humans was also predicted according to the “Committee for Medicinal Products for Human Use. Guideline on Strategies to Identify and Mitigate Risks for First-in-Human Clinical Trials with Investigational Medical Products” [29]. The human PK parameters were predicted using different allometric approaches based on two-species allometry (rat and monkey) and single-species allometry [19].

Complement activation by systemically administered allometric scaling ASOs in humans is believed to be dependent upon C_max_ [22]. For some antisense drugs, the C_max_ correlates across nonclinical species with the mg/kg dose, and in such instances mg/kg scaling would be justified [30]. An alternative HED estimation approach can be derived from estimation of the Pharmacodynamic Active Dose (PAD).

The dose showing efficacy in the mouse animal model was 25 mg/kg [2,4]. This dose has been shown to be safe, but may not correspond to the NOAEL following i.v. administration for the mouse. However, once the maximum recommended safe starting dose (MRSD) has been determined via the NOAEL approach described above (Equation (2)), it may be of value to compare it to the PAD derived from an appropriate pharmacodynamic model. As the PAD is from an in vivo study, a HED can be derived from a PAD estimate using a body surface area converting factor (BSA-CF) [20]. The human PAD value based on mouse effective dose and mouse BSA-CF can be expressed as:human PAD = 25 mg/kg / (70/0.025)^(1 − 0.67)^ = 1.82 mg/kg (5)

This HED value should be compared directly to the MRSD. In practice, the MRSD for the clinical trial should be determined by dividing the HED derived from the animal NOAEL by the safety factor. The default safety factor that should normally be used is 10. If the human PAD (the pharmacologic HED) is lower than the MRSD, it may be appropriate to decrease the clinical starting dose for pragmatic or scientific reasons. This is not the case in our study, as the predicted PAD was slightly higher than the MRSD value even before applying the additional safety factor.

In the absence of a NOAEL in monkey, the rat NOAEL was used as it was the only available NOAEL value. As a more conservative approach, the HED was calculated based on mg/m^2^, which is considered to be a method suitable for predicting lower dose values (than when scaling directly on the mg/kg base). In this case a HED of 0.78 mg/kg was predicted versus 5.0 mg/kg NOAEL, which would have been the value if the scaling had been directly based on the mg/kg normalization.

An alternative HED estimation approach can be derived from a PAD estimate. In this case, the pharmacological HED calculated was slightly higher than the MRSD value: 1.8 vs. 0.78 mg/kg/day. Conversely, the exposure in rats at the NOAEL dose (after 18 days of treatment) was
 AUC_0–t_ (h·ng/mL) = 10,398 in females and 12,157 in males.(6)

Table 6 summarizes all the human CL values predicted using the different methods described above.

Finally, these results were integrated into multiple allometric interspecies scaling approaches which were used to draw inferences about LNA-i-miR-221 PK and safe dose in humans in the absence of other data in humans (Table 7).

The calculated plasma exposure in humans, based on the LNA-i-miR-221 estimated human CL value (geometric mean of the different allometric scaling approaches) and on the NOAEL dose in rats, scaled to humans according to Equation (2), was 9849 h·ng/mL using CLp and 9321 h·ng/mL using CLp^u^. According to the different allometric scaling approaches used, the predicted human exposure ranged from 6741 to 12,686 h·ng/mL based on total CL uncorrected for plasma protein binding, and from 6044 to 12,781 h·ng/mL based on unbound CL. The measured exposure (AUC) in rats at the NOAEL and the predicted human exposure at the HED seemed then to be similar.

For these allometric approaches we excluded PK data from the mouse study because the contribution of the initial distribution phase to the total AUC was probably underestimated, considering the absence of data for the first half hour after infusion. The predicted values, based on rat and monkey PK data, were similar and coherent. Considering the uncertainty about the mouse PK parameters, the effective AUC in the mouse efficacy model at the effective dose of 25 mg/kg was approximately 3226 h·ng/mL, a value around 2–3-fold lower than the exposure in humans at the HED (calculated using the measured rat NOAEL level before applying any corrective safety factor). Conversely the PAD, scaled according to the approach suggested by the FDA guidelines, estimated an effective human dose approximately 2-fold higher than the HED, but still in the same order of magnitude.

## 3. Methods

### 3.1. Chemicals and Materials

The study was approved by the Citoxlab France Ethical Committee (CIC48). LNA-i-miR-221 was provided by BioSpring GmbH (Frankfurt, Germany). Pri-miR-221-222 (nucleotide sequence (5′ > 3′): GAG-AAT-GAA-AAA-TCG) was used as internal standard for the bioanalytical method and was provided by Exiqon (Vedbaek, Denmark). The 17-mer long single-strand oligonucleotide (nucleotide sequence (5′ > 3′): ACA-GAC-AGA-TGT-ATG-CA) used to saturate the non-specific absorption on the surface of ultrafiltration device was purchased from Eurofins Genomics. Formic acid (code 42375500) was purchased from Acros Organics. Ammonium hydroxide (code 221228) was purchased from Fluka. The phenol chloroform:isoamyl alcohol (25:24:1 v:v:v extraction solvent; product code P2069-400 mL), hexafluoroiso-propyl alcohol (code 105228), triethylamine (code STBH2222), and phosphate-buffered saline (PBS) (code P447-100TAB) were purchased form Sigma-Aldrich. Methanol (code 451230000) was obtained from Carlo Erba. The Amicon Ultra-0.5 centrifugal filter, 30 kD ultrafiltration units (30 kDa cut-off, low-binding regenerated cellulose membrane) were purchased from Merck Millipore. About 50 mL of pooled frozen plasma from human, monkey, and rat (pool of donors) obtained from blood treated with K_2_EDTA was used. Information are reported in Table 9. The PK data were collected by sparse sampling (mouse) or as individual subject PK (monkey), meaning that a single profile and a single set of PK parameters was collected for these species and no statistic was applicable. For rats, individual PK parameters and the relative statistics are reported in Table 10.

### 3.2. Ultrafiltration Procedure

#### 3.2.1. LNA-i-miR-221 Protein Binding in Human, Rat, and Monkey

To limit non-specific binding of the LNA-i-miR-221 onto the Amicon vials, they were coated with a 17-mer long single-strand custom-synthesized oligonucleotide. A 500 µL aliquot of coating working solution (10 µM) was added to each Amicon filter, which was capped and kept at room temperature for 30 min. The Amicon units were then centrifuged at 14,000 rpm for 10 min at room temperature, after which the insert was removed and the vials were vortexed for 1 min to coat the inner walls.

Human, monkey, and rat plasma samples were spiked with the LNA-i-miR-221 at 1.0 and 10 µM (final concentrations). After brief vortexing, the samples were incubated in a thermostatic shaking plate at 37 °C for 30 min to allow the test item to bind to the plasma proteins and reach equilibrium. Afterwards, for each LNA-i-miR-221 concentration tested, a 40 µL plasma aliquot was removed and added to a 360 µL aliquot of PBS (*n* = 3). A 500 µL spiked plasma aliquot was loaded into the coated Amicon ultrafiltration vials (*n* = 3). The vials were centrifuged at 14,000 rpm for 10 min at room temperature. After removing the insert, a 360 µL aliquot of filtrate was transferred into an Eppendorf vial containing 40 µL of human, monkey, or rat plasma. All samples were processed in the same way as the calibration curve samples. The percentage of unbound LNA-i-miR-221 was calculated as the ratio between its concentration in the filtered extract and its concentration in the non-filtered extract multiplied by 100. The percentage of protein-bound LNA-i-miR-221 was calculated by subtracting the unbound fraction from 100. Propranolol was used as a reference control. The incubations were performed using the Amicon ultrafiltration vials as for the LNA-i-miR-221, using a 1 µM concentration.

#### 3.2.2. Non-Specific Binding of LNA-i-miR-221 to Amicon Plastic

PBS was spiked with LNA-i-miR-221 (1.0 and 10 µM, final concentrations) in a glass tube. For each LNA-i-miR-221 concentration, a 360 µL aliquot was removed and added to a 40 µL aliquot of rat plasma (samples representing 100% extraction efficiency; *n* = 3). A second 500 µL aliquot was loaded into the Amicon ultrafiltration vials (*n* = 3). The vials were centrifuged at 14,000 rpm for 10 min at room temperature. After removing the insert, 360 µL of filtrate was transferred into an Eppendorf vial and added to a 40 µL aliquot of rat plasma. All samples were then processed in the same way as the calibration curve samples. The percentage of test compound non-specific binding was calculated as the ratio between the test item concentration in the filtered extract and its concentration in the non-filtered extract, multiplied by 100 [17].

### 3.3. Analytical Conditions

The bioanalytical method adopted was based on a previously published method [8], but with minor differences as summarized here. The LNA-i-miR-221 quantification method was based on liquid chromatography coupled with mass spectrometry. The ratios of LNA-i-miR-221 concentration in human, rat, or monkey plasma to PBS (10/90 *v*/*v*) were calculated over the range of 10–2000 ng/ml, based on the calibration curve fitted by the weighted linear regression function (1/x2), with at least 75% of back-calculated concentrations within 15% accuracy. Analyte detection was done using an Orbitrap Q-Exactive (Thermo Scientific) apparatus operating in negative electrospray ionization mode. The HPLC separation was performed using a Halo C18 column (50 × 2.1 mm, 2.7 µm—Advanced Materials Technology, Inc.). Analytes were eluted by applying a flow gradient separation (at 400 µL/minute), starting from 80% of phase A and 20% of phase B (where A was Milli-Q water/Hexafluoroisopropanol (HFIP)/trimethylamine (TEA) 100/4/0.2 *v*/*v*/*v* and B was acetonitrile/HFIP/TEA 100/4/0.2 *v*/*v*/*v*) and increasing the percentage of B up to 80% over 2 min. The composition of mobile phase was then kept constant for an additional 1.6 min and then set back to the initial conditions, leaving the column to re-equilibrate for 2.4 min before a new sample injection. The mass spectrometer was operated in full MS scan between 700 and 900 *m/z* and was set at 70,000 FWHM mass resolution power. LNA-i-miR-221 was detected following the *m/z* 875.0175 signal. Its retention time was approximately 2.65 min, while the internal standard eluted at approximately 2.75 min. A 400 µL sample matrix of plasma/PBS 10/90 *v*/*v* was extracted with 100 µL of phenol/chloroform/isoamyl alcohol 25/24/1 *v*/*v*/*v* solvent. The organic fraction was collected and evaporated under heated nitrogen gas (at 35 °C). After evaporation, the processed sample was re-suspended using 80 µL of the mobile phase mix (A/B = 80/20) before LC-MS analysis.

## 4. Conclusions

Definition of an initial dose for first-in-human studies is complex, and a case-by-case approach may be appropriate depending on the product category (i.e., new chemical entities (NCE) or new biological entities (NBE)). A conservative and consistent approach is required because safety is the most important aspect from a regulatory perspective. For LNA–oligonucleotides, few examples of human PK prediction are available in the literature. The objective of our investigation was to apply and compare different allometric approaches for predicting the CL and exposure of the novel drug candidate LNA-i-miR-221 in humans and to draw inferences about safe human plasma levels in the absence of human data. This was accomplished by applying multiple allometric interspecies scaling approaches to predict the CL of LNA-i-miR-221 oligonucleotides in humans from animal data. In the absence of monkey NOAEL values, the rat NOAEL of LNA-i-miR-221 was used. The HED was calculated as mg/m^2^ and the human equivalent dose (corresponding to the NOAEL in rats) was predicted to be 0.78 mg/kg/day. The calculated plasma exposure in humans, based on the estimated human LNA-i-miR-221 CL value, is 9849 h·ng/mL using CLp and 9321 h·ng/mL using CLp^u^. According to the different allometric scaling approaches used, the predicted exposure in humans ranges from 6741 to 12,686 h·ng/mL when based on total CL uncorrected for plasma protein binding, and from 6044 to 12,781 h·ng/mL when based on unbound CL. Finally, the human extrapolated PAD value based on the mouse effective dose and mouse BSA-CF was 1.82 mg/kg. The project flow and the results are summarized in Figure 4.

The optimal PK of LNA-i-miR-221 relies on a large systemic Vd and broad tissue penetration. The plasma protein binding offers the additional advantage of preventing rapid renal clearance. Our evidence that all tested species showed similar unbound fraction values is reassuring in terms of expected similar interspecies PK behavior, which is a basic assumption when allometric extrapolation is used to predict human CL. The NOAEL and the protein binding results led us to predict the safe HED and the relative exposure in humans, and to provide further valuable information on LNA-i-miR-221 PK to be confirmed in the ongoing first-in-human study.

## 5. Patents

LNA-i-miR-221 and its uses are patented: European Patent No. 2943570; United States Patent No. US 9,404,111,B2; Italian Patent No. 0001429326.

## Figures and Tables

**Figure 1 cancers-12-00027-f001:**
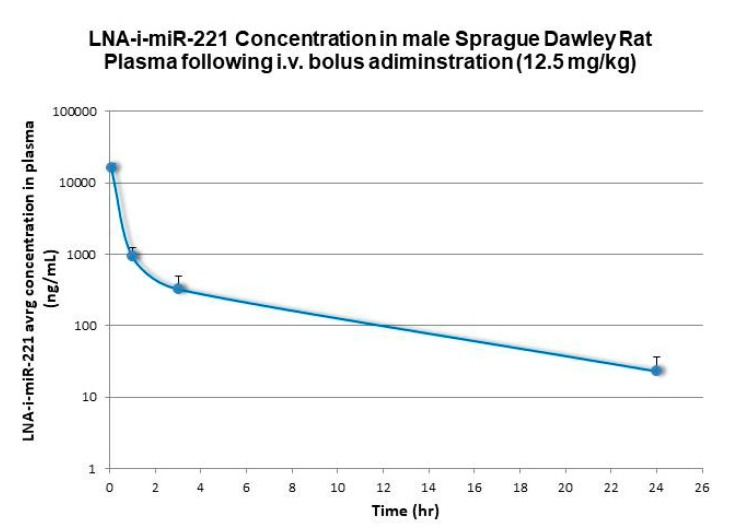
plasma concentration *versus* time profiles (sparse sampling) following single intravenous (bolus) administration at 12.5 mg/kg to male and female Sprague-Dawley rats (semi logarithmic scale). Error bars represent standard deviations (*n* = 3).

**Figure 2 cancers-12-00027-f002:**
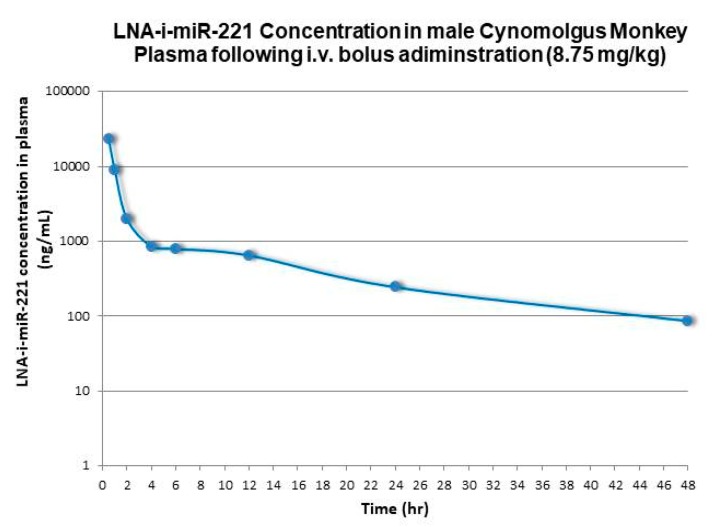
plasma concentration *versus* time profiles following single intravenous (bolus) administration at 8.75 mg/kg to male Cynomolgus Monkey (semi logarithmic scale).

**Figure 3 cancers-12-00027-f003:**
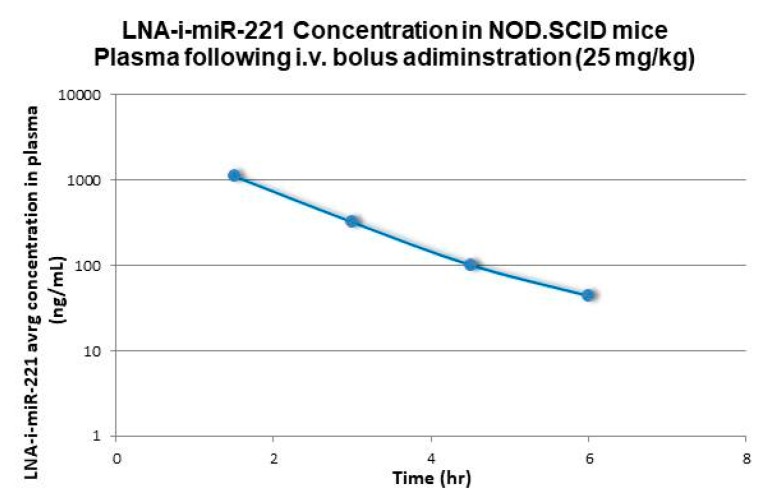
plasma concentration *versus* time profiles (sparse sampling) following single intravenous (bolus) administration at 25 mg/kg to NOD.SCID mice (semi logarithmic scale). Plasma concentrations at 12h and 24h were below the limit of quantification (LLQ = 25ng/mL). Mouse PK was not used for the allometric extrapolation (see main text for details).

**Figure 4 cancers-12-00027-f004:**
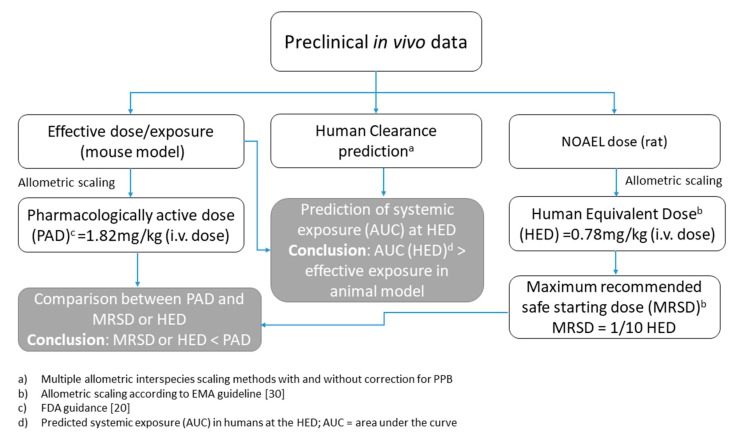
Schematic representation of the methodological approaches and results discussed in the work. Arrows connect the input information used to estimate the corresponding output parameter; references to the applied methods are also cited. Based on the estimated parameters, a short conclusion is reported. Additional details are described in the main text.

**Table 1 cancers-12-00027-t001:** Pharmacokinetic (PK) parameters measured in preclinical species after i.v. bolus administration of a novel phosphorothioate backbone 13-mer locked nucleic acid oligonucleotide-targeting miRNA-221 (LNA-i-miR-221).

Species	Dose (i.v. Bolus)	C_0_/Dose	AUC ^1^	AUC/Dose	CL	Terminal Half-Life	Vd Terminal
	mg/kg	ng/mL/mg	(h·ng/mL)	(h·ng/mL)/mg	mL/min/kg	h	mL/Kg
Rat	12.5	4462	27192	2160	7.9	3.9	2626
Mouse ^2^	25	82	3226	129	129	1.5	16443
Monkey	8.75	7017	49300	5634	3.0	12.8	3286

^1^ for mouse and rat AUC was the corresponding AUC_0–∞_ after a single dose; for monkey, the AUC value corresponds to AUC_0–48 h_ after single dose. ^2^ mouse PK parameters here reported should be considered only as indicative.

**Table 2 cancers-12-00027-t002:** Average body weights (kg) used for allometric scaling.

Species	Average Body Weights (kg) Used for Allometric Scaling
Mouse	0.025
Rat	0.25
Monkey (Cynomolgus)	3.75
Human	70

The values have not been adjusted for the real animal weights.

**Table 3 cancers-12-00027-t003:** Two-species direct allometric scaling, based on rat and monkey total clearance.

b	−0.362198
a	0.6789
CL_human_ (mL/min/kg)	1.02
CL_human_ (mL/min)	71.7

**Table 4 cancers-12-00027-t004:** Two-species allometric scaling according to Equation (3) (Tang method, 2007), based on rat and monkey total clearance.

b	0.65
a_rat–monkey_	0.6789
CL_human_ (mL/min/kg)	1.1
CL_human_ (mL/min)	76

**Table 5 cancers-12-00027-t005:** One-species allometric scaling according to Equation (4).

Species Used for Scaling	Human CLp (mL/min/kg)	Human CLp (mL/min)
	Single Species Allometry	
rat	1.9	135
monkey	1.4	100

**Table 6 cancers-12-00027-t006:** Predicted LNA-i-miR-221 human PK parameters based on total CLp using different allometric approaches.

	Total Plasma Clearance	AUC/Dose	AUC for 0.78 ^‡^ mg/kg i.v. Dose	AUC for 1.82 ^‡^ mg/kg i.v. Dose	AUC for 5.0 ^‡^ mg/kg i.v. Dose
Allometric Method Used for Prediction	mL/min/kg	h·ng/mL·(mg Dose)^−1^	h·ng/mL	h·ng/mL	h·ng/mL
Direct scaling two-species (r, mk) (Equation (1a))	1.0	16,264	12,686	29,600	81,318
Tang et al. method two-species (r, mk) (Equation (3))	1.1	15,443	12,046	28,106	77,215
One-species (r) allometric scaling (Equation (4))	1.9	8643	6741	15,730	43,214
One-species (mk) allometric scaling (Equation (4))	1.4	11,711	9135	21,315	58,557
	**Geometric mean**	**Geometric** **mean**	**Geometric** **mean**	**Geometric** **mean**	**Geometric** **mean**
	**1.3**	**12,627**	**9849**	**22,981**	**63,135**

^‡^ HED predicted according to Equation (2).

**Table 7 cancers-12-00027-t007:** Predicted LNA-i-miR-221 human PK parameters based on unbound CLp^u^ using different allometric approaches.

	Total Plasma Clearance	AUC/Dose	AUC for 0.78 ^‡^ mg/kg i.v. Dose	AUC for 1.82 ^‡^ mg/kg i.v. Dose	AUC for 5.0 ^‡^ mg/kg i.v. Dose
Allometric Method Used for Prediction	mL/min/kg	h·ng/mL·(mg Dose)^−1^	h·ng/mL	h·ng/mL	h·ng/mL
Direct scaling two-species (r, mk) (Equation (1a))	1.0	16,386	12781	29,823	81,931
Tang et al. method two-species (r, mk) (Equation (3))	1.15	14,462	11280	26,320	72,309
One-species (r) allometric scaling (Equation (4))	2.2	7749	6044	14,103	38,743
One-species (mk) allometric scaling (Equation (4))	1.5	11,106	8662	20,212	55,528
	**Geometric mean**	**Geometric** **mean**	**Geometric mean**	**Geometric mean**	**Geometric mean**
	**1.4**	**11950**	**9321**	**21,749**	**59,750**

M = mouse, r = rat, mk = monkey; ^‡^ HED predicted according to Equation (2); human PAD predicted according to Equation (5); assuming hum NOAEL = rat NOAEL.

**Table 8 cancers-12-00027-t008:** Protein binding values for the test item in human, monkey and rat plasma. Data are presented as mean ± standard deviation (*n* = 3). Unbound fraction (f_u_) values (averages of the two tested concentrations) used for the unbound clearance estimation.

Species	LNA-i-miR-221 Concentration			
	1 µM	10 µM	Mean PPB	Mean f_u_
Human	98.6 ± 0.32	98.5 ± 0.09	98.55	0.0145
Monkey	98.2 ± 0.39	99.05 ± 0.39	98.63	0.0138
Rat	98.5 ± 0.17	98.9 ± 0.17	98.70	0.0130

PPB = plasma protein binding; mean f_u_ = (100 − mean PPB%)/100.

**Table 9 cancers-12-00027-t009:** Blank plasma used in the study.

Species	Sex	Supplier	Anticoagulant
Rat ^a^ (Sprague–Dawley)	Male Pool of 25 animals	Citoxlab	K_2_EDTA
Monkey ^b^ (Cynomolgus)	Male Pool of 10 animals	Citoxlab	K_2_EDTA
Human ^c^	Male Pool of 10 donors	Biopredic (PLA152A050)	K_2_EDTA

^a^. (animal age: range 9–12 weeks, median 10 weeks) (animals’ body weight not recorded); ^b^. the animals were 33 to 58 months old (median 44 months) and they had a mean body weight of 4.1 kg (range: 3.2 kg to 4.8 kg); ^c^. (all Caucasians, aged 32–64 years, median 48 years).

**Table 10 cancers-12-00027-t010:** Rats’ individual PK parameters and the relative statistics.

Sex		λ	t_1/2_	C_0_	AUC_0–∞_	AUC_0–∞_/Dose	AUC Extrapolated	Vz	Cl
		1/h	h	ng/mL	h·ng/mL		%	mL/kg	mL/min/kg
**F**		0.174	3.99	43,094	23,980	1918	0.869	3000	8.7
		0.187	3.70	47,033	24,961	1997	0.603	2672	8.3
		0.187	3.70	64,272	31,455	2516	0.561	2120	6.6
	**mean**	0.183	3.80	51,467	26,799	2144	0.678	2597	7.9
	**SD**	0.008	0.168	11,264	4062	325	0.167	445	1.1
	**CV%**	4	4	22	15	15	25	17	0.2
**M**		0.178	3.90	49,251	24,364	1949	0.724	2890	8.6
		0.184	3.76	86,432	34,091	2727	0.490	1990	6.1
		0.175	3.95	44,593	23,120	1850	0.792	3083	9.0
	**mean**	0.179	3.87	60,092	27,192	2175	0.669	2654	7.9
	**SD**	0.005	0.099	22,930	6008	481	0.158	583	1.6
	**CV%**	3	3	38	22	22	24	22	0.3
**M + F**	**mean**	0.181	3.83	55,779	26,995	2160	0.673	2626	7.9
	**SD**	0.006	0.131	16,834	4592	367	0.145	465	1.2
	**CV%**	3	3	30	17	17	22	18	0.0

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
