# Peer review of "Allometric Scaling Approaches for Predicting Human Pharmacokinetic of a Locked Nucleic Acid Oligonucleotide Targeting Cancer-Associated miR-221"

_cancers, 2019, doi:10.3390/cancers12010027_

Round 1

Reviewer 1 Report

This study deals with the pharmacokinetics properties of LNA-i-miR-221 targeting microRNA-221, including unbound/total clearance. The authors investigated the LNA-i-miR-221 protein binding in rat, monkey and human, and they generated an ultrafiltration method to study the binding of LNA-i-miR-221 to plasma proteins. They also showed that a pharmacokinetic advantage by preventing rapid renal clearance was caused by plasma protein binding of LNA-i-miR-221. This alas is not the best journal for this paper. The manuscript is very concerned with the pharmacokinetics properties and a journal concerned with that would be the far better option. I am sure you will have no trouble getting this published in such a journal.

Author Response

We well understand the concern made by the Rewiever . However, we believe that data included in our study on the pharmacokinetics properties of a novel LNA-miR-221 actually provides novel information on a first-in class anti-cancer therapeutic that is presently in early clinical investigation, which has been designed on the results here reported. To underline the relevance of the targeting of miR-221 in human cancer, where this miRNA is highly deregulated, we slightly modified the title.

Reviewer 2 Report

Here the authors reported an ultrafiltration method to determine pharmacokinetics properties of LNA-i-miR-221, including unbound/total clearance, protein binding in three different species including rat, monkey and human. This work is interesting and novel.

I have only two comments: 1. Please provide the approval information from Ethics Committee.2. Please clarify the information of “rat, monkey and human”, eg: gender/age…3. Data are recommended to been shown as “Mean±SD”, if possible.

Author Response

1: Please provide the approval information from Ethics Committee

The animals studies were conducted in compliance with Animal Health regulations, in particular Council Directive No. 2010/63/EU of 22 September 2010 on the protection of animals used for scientific purposes. The Citoxlab France Ethical Committee (CEC) – accredited by the International Association for Assessment and Accreditation of Laboratory Animal Care (AAALAC) – reviewed the relative study protocols for all the “in life” phases mentioned in this work, in order to assess the compliance with the corresponding authorized “project” as defined in the Directive 2010/63/EU. The CEC notified the Study Director of the review before the finalization of the study plan. During the study, the CEC was regularly informed of any amendments to the study plan which could have an impact on animal welfare. We add a short sentence in the methods section of the revised manuscript.

2: Please clarify the information of “rat, monkey and human”, eg: gender/age…

Animal information are added to the revised manuscript and are reported in tables 9 and 10, and in methods section.

3: Data are recommended to been shown as “Mean±SD”, if possible

As suggested by Reviewer, we included in the revised manuscript the PK animal data. These data have been collected as sparse sampling (for mouse) or as individual animals PK (for monkey). This means that a single profile and a single set of PK parameters were collected for these species and no statistics could be applied. For Rats, individual PK parameters and the relative statistics are reported in the tab. 10 of the revised manuscript.

Reviewer 3 Report

The authors have described how they designed allosteric scaling modality for predicting human PK of a Locked nucleic acid oligonucleotide targeting miR-221.

As mentioned by the authors, miR221 has been implicated in various diseases and is an attractive target to mitigate those diseases. LNA is a viable option. 

The authors have compared the clearance of the oligonucleotide in mice, rat and monkey, which are reasonable models for a novel drug PK study. The results are interesting and provide a rationale for using this modality for dose-finding strategies for first-in-human clinical trials, where there is a lack of human data.

However, in order to be considered for publication, I propose:

A graph showing plasma concentrations of LNA on Y axis with time as X axis would be great. Please elaborate the PK analysis results and simplify the explanation. I do not understand the last two references (29 and 30). How does it relate to the context ? The manuscript has some serious grammatical flaws and typos that need to be corrected.

Author Response

1: A graph showing plasma concentrations of LNA on Y axis with time as X axis would be great

As suggested by the Reviewer, we included in the revised manuscript graphs showing plasma concentrations of LNA-i-miR-221 versus time (see fig 1-3) for rat, monkey and mouse. The simulation of the PK plasma profile with time in man requires the estimation of compartmental PK parameters in humans. In our work non-compartmental PK analysis has been applied in preclinical species, as normally required for Regulatory toxicokinetics analysis. Extrapolation of non-compartmental PK parameters in humans from these data allowed us the estimation of the human equivalent dose (HED) and the Pharmacodynamic Active Dose (PAD), which was the main scope of the current work. The possibility of scaling allometrically the compartmental parameters in humans has been indeed considered. We planned to collect enough PK data from the Fist-in-Human study to evaluate the best estimation approach and then directly compare the predicted plasma profile by our non- compartmental approach and the measured profile in humans, to be described in a follow up study focused on human PK data.

2: Please elaborate the PK analysis results and simplify the explanation.

We implemented the manuscript following the Reviewer suggestion and we added three figure with the PK data and discussed the results at pag 3 in the revised manuscript.

3: I do not understand the last two references (29 and 30). How does it relate to the context ?

We apologize for the mistakes occurred during the process of editing. We included corrected references.

4: The manuscript has some serious grammatical flaws and typos that need to be corrected.

As suggested, we corrected typos and we made the major effort to improve writing and editing.

Round 2

Reviewer 1 Report

The manuscript has been very concerned with the pharmacokinetics properties and revised according to other reviewers’ comments. In my opinion, your analysis and results are adequate and no problems, and I understand that your results have been useful for PK/PD study of LNA oligonucleotides. Some of your explanations difficult to follow for not specialists, if possible, it is better to indicate overview of your results by, for example, simplified schema, like graphical abstract for understanding of readers.

Author Response

As suggested by the reviwer we included an additional figure that recapitulate methodological approches and results of the work

Reviewer 3 Report

The authors have addressed the concerns adequately.

Author Response

Thank you for yours suggestions